# Mixture-of-Experts Variational Autoencoder for clustering and generating from similarity-based representations on single cell data

**Andreas Kopf**[1,5], **Vincent Fortuin**[2,4], **Vignesh Ram Somnath**[1], **Manfred Claassen**[3]*

**1** Institute of Molecular Systems Biology, Department of Biology, ETH Zürich, Zurich, Switzerland, **2** Biomedical Informatics Group, Department of Computer Science, ETH Zürich, Zurich, Switzerland, **3** Division of Clinical Bioinformatics, Department of Internal Medicine I, University of Tübingen, Tübingen, Germany, **4** Swiss Institute of Bioinformatics (SIB), Zurich, Switzerland, **5** Life Science Graduate School Zurich, PhD Program Systems Biology, Zurich, Switzerland

* manfred.claassen@med.uni-tuebingen.de

**Data Availability Statement:** All relevant data are within the paper and its Supporting information files. MoE-Sim-VAE is available at the following

## Abstract

Clustering high-dimensional data, such as images or biological measurements, is a long-standing problem and has been studied extensively. Recently, Deep Clustering has gained popularity due to its flexibility in fitting the specific peculiarities of complex data. Here we introduce the Mixture-of-Experts Similarity Variational Autoencoder (MoE-Sim-VAE), a novel generative clustering model. The model can learn multi-modal distributions of high-dimensional data and use these to generate realistic data with high efficacy and efficiency. MoE-Sim-VAE is based on a Variational Autoencoder (VAE), where the decoder consists of a Mixture-of-Experts (MoE) architecture. This specific architecture allows for various modes of the data to be automatically learned by means of the experts. Additionally, we encourage the lower dimensional latent representation of our model to follow a Gaussian mixture distribution and to accurately represent the similarities between the data points. We assess the performance of our model on the MNIST benchmark data set and challenging real-world tasks of clustering mouse organs from single-cell RNA-sequencing measurements and defining cell subpopulations from mass cytometry (CyTOF) measurements on hundreds of different datasets. MoE-Sim-VAE exhibits superior clustering performance on all these tasks in comparison to the baselines as well as competitor methods.

## Author summary

Clustering single cell measurements into relevant biological phenotypes, such as cell types or tissue types, is an important task in computational biology. We developed a computational approach which allows incorporating prior knowledge about the single cell similarity into the training process, and ultimately achieve significant better clustering performance compared to baseline methods. This single cell similarity can be defined to benefit specific needs of the modeling goal, for example to either cluster cell type or tissue type, respectively.

In addition, we are able to generate new realistic single cell data from a respective mode of the phenotype due to the architecture of the model, which consists of smaller

Github repository: https://github.com/andkopf/MoESimVAE.

**Funding:** AK is supported by the "SystemsX.ch HDL-X" and "ERASysApp Rootbook" and PHRT 2017-103. VF is supported by a PhD fellowship from the Swiss Data Science Center and by the PHRT grant #2017-110 of the ETH Domain. The funders had no role in study design, data collection and analysis, decision to publish, or preparation of the manuscript.

**Competing interests:** The authors have declared that no competing interests exist.

sub-models learning the different modes of the data. Compared to competitor methods, we show significantly better results on clustering and generation of handwritten digits of the MNIST data set, on clustering seven different mouse organs from single-cell RNA sequencing measurements, and on clustering cell types in over 272 different datasets of Peripheral Blood Mononuclear Cell measured via CyTOF.

This is a *PLOS Computational Biology* Methods paper.

## Introduction

Clustering has been studied extensively [1, 2] in machine learning and has found wide application in identifying grouping structure in high dimensional biological data such as various omics data modalities. Recently, many Deep Clustering approaches were proposed, which modified (Variational) Autoencoder ((V)AE) architectures [2, 3] or by varying regularization of the latent representation [4–7].

The reconstruction error usually drives the definition of the latent representation learned from an AE or VAE. The representation for AE models is unconstrained and typically places data objects close to each other according to an implicit similarity measure that also yields favorable reconstruction error. In contrast, VAE models regularize the latent representation such that the represented inputs follow a certain variational distribution. This construction enables sampling from the latent representation and data generation via the decoder of a VAE. Typically, the variational distribution is assumed standard Gaussian, but for example Jiang *et al.* [7] introduced a mixture-of-Gaussians variational distribution for clustering purposes.

A key component of clustering approaches is the choice of similarity metric for the considered data objects which we try to group [8]. Such similarity metrics are either defined *a priori* or learned from the data to specifically solve classification tasks via a Siamese network architecture [9]. Dimensionality reduction approaches, such as UMAP [10] or t-SNE [11], allow to specify a similarity metric for projection and thereby define the data separation in the inferred latent representation.

In this work, we introduce the Mixture-of-Experts Similarity Variational Autoencoder (MoE-Sim-VAE), a new deep architecture that performs similarity-based representation learning, clustering of the data and generation of data from each specific data mode. Due to a combined loss function, it can be jointly optimized. We empirically assess the scope of the model and present superior clustering performance on the canonical benchmark MNIST. Moreover, in an ablation study, we show the efficiency and precision of MoE-Sim-VAE for data generation purposes in comparison to the most related state-of-the-art method [7]. We achieve superior results on the identification of tissue- or cell type groupings via MoE-Sim-VAE on a murine single-cell RNA-sequencing atlas and mass cytometry measurements of Peripheral Blood Mononuclear Cells.

## Materials and methods

### MoE-Sim-VAE

Here we introduce the Mixture-of-Experts Similarity Variational Autoencoder (MoE-Sim-VAE, Fig 1). The model is based on the Variational Autoencoder [12]. While the encoder network is shared across all data points, the decoder of the MoE-Sim-VAE consists of a number

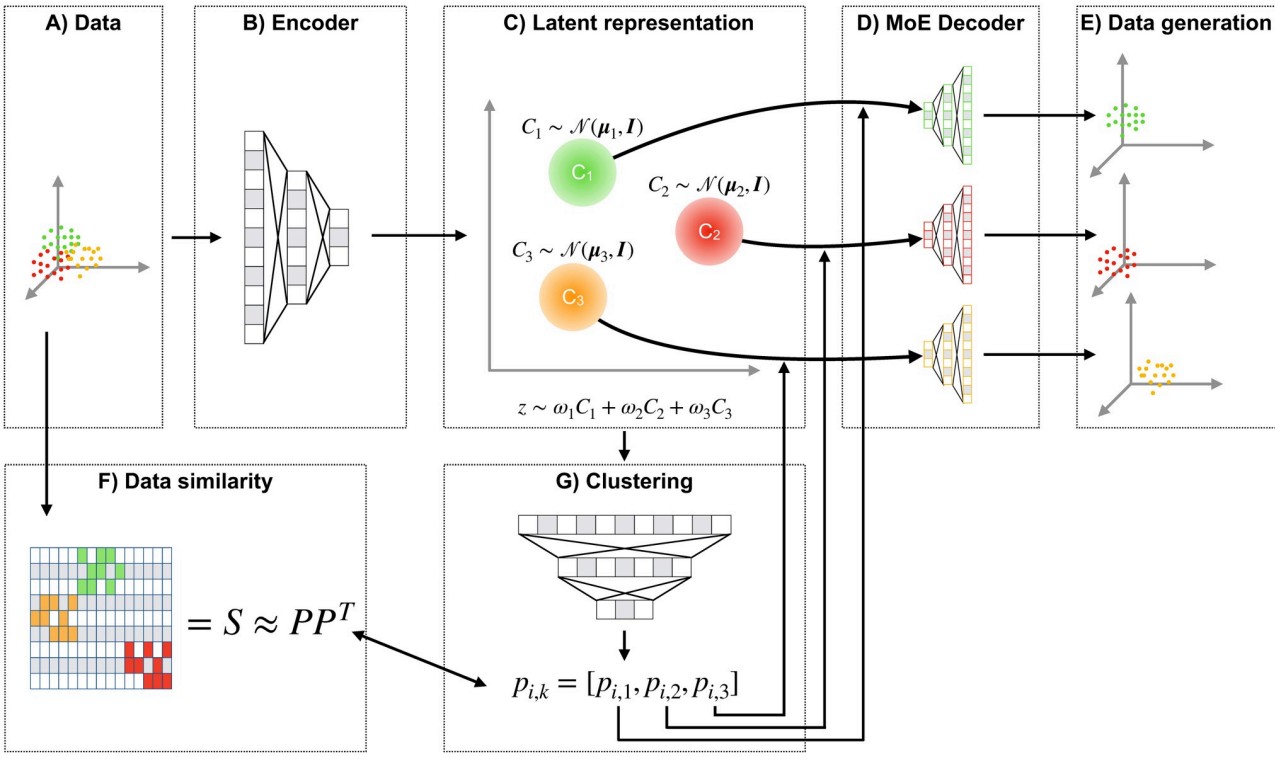

**Fig 1. Schematic overview of MoE-Sim-VAE.** Data (in panel A) gets encoded via an encoder network (B) into a latent representation (C) which is trained to be a mixture of standard Gaussians. Via a clustering network (G), which is trained to reconstruct a user-defined similarity matrix (F), the encoded samples get assigned to the data mode-specific decoder subnetwork (which we call experts) in the MoE Decoder (D). The experts reconstruct the original input data and can be used for data generation when sampling from the variational distribution (E).

of $K$ different subnetworks, forming a Mixture-of-Experts architecture [13]. Each subnetwork constitutes a generator for a specific data mode and is learned from the data.

The variational distribution over the latent representation is defined to be a mixture of multivariate Gaussians, first introduced by Jiang *et al.* [7]. In our model, we aim to learn the mixture components in the latent representation to be standard Gaussians

$$z \sim \sum_{k=0}^{K} \omega_k \mathcal{N}(\boldsymbol{\mu}_k, \boldsymbol{I}) \tag{1}$$

where $\omega_k$ are mixture coefficients, $\boldsymbol{\mu}_k$ are the means for each mixture component, $\boldsymbol{I}$ is the identity matrix and $K$ is the number of mixture components. The dimension of the latent representation $z$ needs to be defined to suit the demands of Gaussian mixtures which have limitations in higher dimensions [14]. Similar to optimizing an Evidence Lower Bound (ELBO), we penalize the latent representation via the reconstruction loss of the data $\mathcal{L}_{reconst}$ and by using the Kullback-Leibler (KL) divergence for multivariate Gaussians [7] on the latent representation

$$\mathcal{L}_{KL} = D_{KL}(\mathcal{N}_0, \mathcal{N}_1) = \frac{1}{2} \{ tr(\boldsymbol{\Sigma}_1^{-1} \boldsymbol{\Sigma}_0) + \\ (\boldsymbol{\mu}_1 - \boldsymbol{\mu}_0)^T \boldsymbol{\Sigma}_1^{-1} (\boldsymbol{\mu}_1 - \boldsymbol{\mu}_0) - k + ln \frac{|\boldsymbol{\Sigma}_1|}{|\boldsymbol{\Sigma}_0|} \} \tag{2}$$

where $k$ is a constant, $\mathcal{N}_0 \sim \mathcal{N}(\boldsymbol{\mu}_0, \boldsymbol{\Sigma}_0 = \boldsymbol{I})$, and $\boldsymbol{I}$ is the identity matrix. Further, $\mathcal{N}_1 \sim \mathcal{N}(\boldsymbol{\mu}_1, \boldsymbol{\Sigma}_1 = diag(\sigma_j))$, where $\sigma_j$ for $j = 1, \ldots, D$, for a number of dimensions $D$, is

estimated from the samples of the latent representation. Finally, we assume $\boldsymbol{\mu}_0 = \boldsymbol{\mu}_1$ resulting in the following simplified objective

$$\mathcal{L}_{KL} = D_{KL}(\mathcal{N}_0, \mathcal{N}_1) = \frac{1}{2}\left\{ tr(\boldsymbol{\Sigma}_1^{-1}\boldsymbol{\Sigma}_0) - k + ln\frac{|\boldsymbol{\Sigma}_1|}{|\boldsymbol{\Sigma}_0|} \right\} \; , \tag{3}$$

to penalize exclusively the covariance of each cluster. It remains to define the reconstruction loss $\mathcal{L}_{reconst}$, where we choose a Binary Cross-Entropy (BCE)

$$\mathcal{L}_{reconst} = \sum_i^N \sum_d^D x_{i,d} \log(x_{i,d}^{reconst}) \tag{4}$$

between the original data $x$ (scaled between 0 and 1) and the reconstructed data $x^{reconst}$, where $i$ iterates the batch size $N$ and $d$ the dimensions of the data $D$. We motivate the BCE loss due to better convergence properties using artificial neural networks in comparison to mean squared error [15]. Finally the loss for the VAE part is defined by

$$\mathcal{L}_{VAE} = \mathcal{L}_{reconst} + \pi_1 \mathcal{L}_{KL} \tag{5}$$

with a weighting coefficient $\pi_1$ which can be optimized as a hyperparameter.

**Similarity clustering and gating of latent representation.** Training of a data mode-specific generator expert requires samples from the same data mode. This necessitates to solve a clustering problem, that is, mapping the data via the latent representation into $K$ clusters, each corresponding to one of the $K$ generator experts. We solve this clustering problem via a clustering network, also referred to as gating network for MoE models. It takes as input the latent representation $z_i$ of sample $i$ and outputs probabilities $p_{ik} \in [0, 1]$ for clustering sample $i$ into cluster $k$. According to this cluster assignment, sample $i$ is then gated to expert $k = \text{argmax}_k \, p_{ik}$ for each sample $i$. We further define the cluster centers $\boldsymbol{\mu}_k$ for $k \in \{1, \ldots, K\}$ similar as in the Expectation Maximization (EM) algorithm for Gaussian Mixture models [16] as

$$\boldsymbol{\mu}_k = \frac{1}{N_k}\sum_{i=1}^N p_{ik}\boldsymbol{z}_i \; , \tag{6}$$

where $N_k$ is the absolute number of data points assigned to cluster $k$ based on highest probability $p_{ik}$ for each sample $i = 1, \ldots, N$. The Gaussian mixture distributed latent representation (via KL loss in Eq 3) is motivation for the empirical computation of the cluster means and further, similar as in the EM algorithm, it allows iterative optimization of the means of the Gaussians. We train the clustering network to reconstruct a data-driven similarity matrix $\boldsymbol{S}$, using the Binary Cross-Entropy

$$\mathcal{L}_{Similarity} = \sum_i^N \sum_j^N \boldsymbol{S}_{i,j} \log((\boldsymbol{PP}^T)_{i,j}) \tag{7}$$

to minimize the error in $\boldsymbol{PP}^T \approx \boldsymbol{S}$, with $\boldsymbol{P} := \{p_{ik}\}_{i\in\{1,\ldots,N\}, k\in\{1,\ldots,K\}}$ where $N$ is the number of samples (e.g., batch size). Intuitively, $\boldsymbol{PP}^T$ approximates the similarity matrix $\boldsymbol{S}$ since values in $\boldsymbol{PP}^T$ are only close to 1 when similar data objects are assigned to the same cluster, similar to the entries in the adjacency similarity matrix $\boldsymbol{S}$. This similarity matrix is derived in an unsupervised way in our experiments (e.g. UMAP projection of the data and k-nearest-neighbors or distance thresholding to define the adjacency matrix for the batch), but can also be used to include weakly-supervised information (e.g., knowledge about diseased vs. non-diseased patients). If labels are available, the model could even be used to derive a latent representation

with supervision. The similarity feature in MoE-Sim-VAE thus allows to include prior knowledge about the best similarity measure on the data.

Moreover, we apply the DEPICT loss from Dizaji *et al.* [4], to improve the robustness of the clustering. For the DEPICT loss, we additionally propagate a noisy probability $\hat{p}_{ik}$ through the clustering network using dropout after each layer. The goal is to predict the same cluster for both, the noisy $\hat{p}_{ik}$ and the clean probability $p_{ik}$ (without applying dropout). Dizaji *et al.* [4] derived as objective function a standard cross-entropy loss

$$\mathcal{L}_{DEPICT} = -\frac{1}{N} \sum_{i=0}^{N} \sum_{k=0}^{K} q_{ik} \log \hat{p}_{ik} \tag{8}$$

whereby $q_{ik}$ is computed via the auxiliary function

$$q_{ik} = \frac{p_{ik}/(\sum_{i'} p_{i'k})^{\frac{1}{2}}}{\sum_{k'} p_{ik'}/(\sum_{i'} p_{i'k'})^{\frac{1}{2}}} \ . \tag{9}$$

We refer to Dizaji *et al.* [4] for the exact derivation. The DEPICT loss encourages the model to learn invariant features from the latent representation for clustering with respect to noise [4]. Looking at it from a different perspective, the loss helps to define a latent representation which has those invariant features to be able to reconstruct the similarity and therefore the clustering correctly. The complete clustering loss function $\mathcal{L}_{Clustering}$ is then defined by

$$\mathcal{L}_{Clustering} = \mathcal{L}_{Similarity} + \pi_2 \mathcal{L}_{DEPICT} \tag{10}$$

with a mixture coefficient $\pi_2$ which can be optimized as a hyperparameter.

**MoE-Sim-VAE loss function.** Finally, the MoE-Sim-VAE model loss is defined by

$$\mathcal{L}_{MoE-Sim-VAE} = \underbrace{\mathcal{L}_{VAE}}_{\mathcal{L}_{reconst}+\pi_1\mathcal{L}_{KL}} + \underbrace{\mathcal{L}_{Clustering}}_{\mathcal{L}_{Similarity}+\pi_2\mathcal{L}_{DEPICT}} \tag{11}$$

which consists of the two main loss functions $\mathcal{L}_{VAE}$, acting as a regularization for the latent representation, and $\mathcal{L}_{Clustering}$, which helps to learn the mixture components based on an *a priori* defined data similarity. The model objective function $\mathcal{L}_{MoE-Sim-VAE}$ can then be optimized end-to-end to train all parts of the model.

## Related work

(V)AEs have been extensively used for clustering [1, 4–6, 17–20]. The most related approaches to MoE-Sim-VAE are Jiang *et al.* [7] and Zhang *et al.* [3].

Jiang *et al.* [7] introduced the VaDE model, comprising a mixture of Gaussians as underlying distribution in the latent representation of a Variational Autoencoder. Optimizing the Evidence Lower Bound (ELBO) of the log-likelihood of the data can be rewritten to optimize the reconstruction loss of the data and KL divergence between the variational posterior and the mixture of Gaussians prior. Jiang *et al.* [7] use two separate networks for reconstruction and the generation process. Further, to effectively generate images from a specific data mode and to increase image quality, the sampled points have to surpass a certain posterior threshold and are otherwise rejected. This leads to an increased computational effort. The MoE Decoder of our model, which is used for both reconstruction and generation, does not need such a threshold.

Zhang *et al.* [3] have introduced a mixture of autoencoders (MIXAE) model. The latent representation of the MIXAE is defined as the concatenation of the latent representation

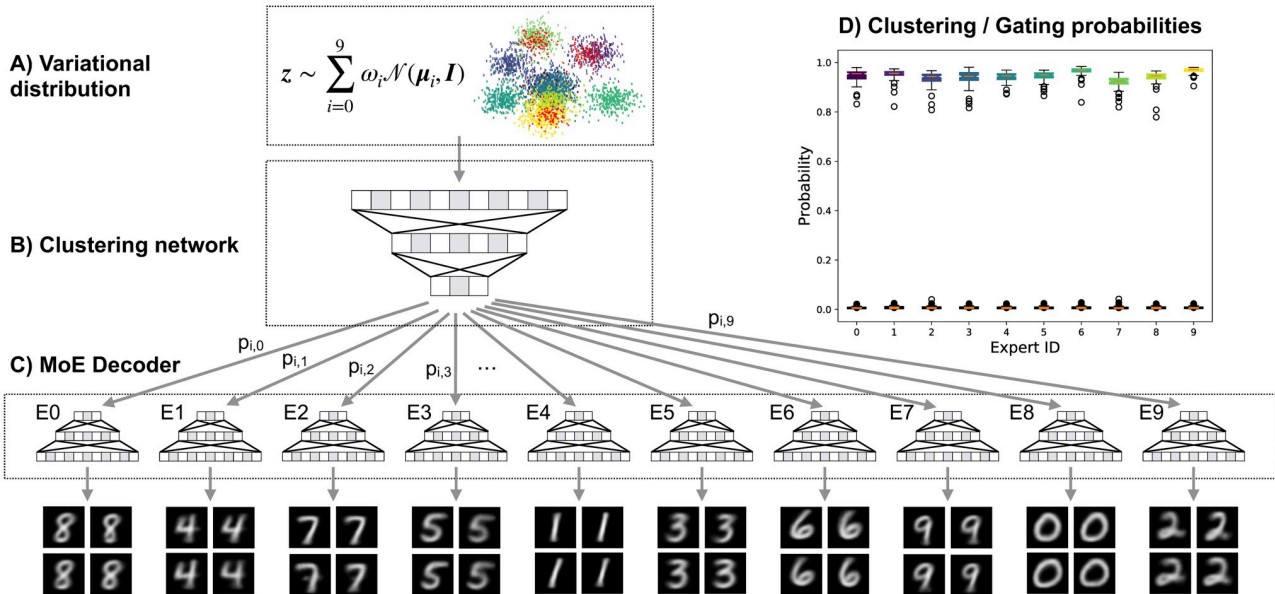

**Fig 2. Generation of MNIST digit images.** Data points from the latent representation were sampled from the variational distribution (A) which is learned to be a mixture of standard Gaussians and then clustered and gated (B) to the data-mode-specific experts of the MoE Decoder (C). (D) All samples from the variational distribution were correctly classified and therefore also correctly gated.

vectors of each single autoencoder in the model. Based on this concatenated latent representation, a Mixture Assignment Network predicts probabilities which are used in the Mixture Aggregation to form the output of the generator network. Each AE model learns the manifold of a specific cluster, similarly to our MoE Decoder. However, MIXAE does not optimize a variational distribution, such that generation of data from a distribution over the latent representation is not possible, in contrast to the MoE-Sim-VAE (Fig 2).

## Results

In the following we report superior clustering and generating results of MoE-Sim-VAE on real world problems. First, we evaluate MoE-Sim-VAE on images from MNIST and show why a MoE decoder is beneficial. Second, we present significantly better clustering results on mouse organ single-cell RNA sequencing data. Third, we apply MoE-Sim-VAE to cluster cell types in Peripheral Blood Mononuclear Cells using CyTOF measurements on 272 distinct data sets significantly better than competitors. (Exact model and optimization details as well as preprocessing steps for all experiments can be found in S1 Text)

### Unsupervised clustering, representation learning and data generation on MNIST

We trained a MoE-Sim-VAE model on images from MNIST. We compared our model against multiple models which were recently reviewed in Aljalbout *et al*. [1], and specifically against VaDE [7] which shares similar properties with MoE-Sim-VAE. The VaDE model is comprising a mixture of Gaussians as underlying distribution in the latent representation of a Variational Autoencoder (more detailed comparison in Section Related work).

**Table 1. Performance comparison of our method MoE-Sim-VAE with several published methods on MNIST.** The table is mainly extracted from [1, 21] and complemented with results of interest. ("-": metric not reported).

| Method | NMI | ACC |
|---|---|---|
| JULE [22] | 0.915 | - |
| CCNN [23] | 0.876 | - |
| DEC [17] | 0.80 | 0.843 |
| DBC [18] | 0.917 | 0.964 |
| DEPICT [4] | 0.916 | 0.965 |
| DCN [5] | 0.81 | 0.83 |
| Neural Clustering [19] | - | 0.966 |
| UMMC [20] | 0.864 | - |
| VaDE [7] | 0.876 | 0.945 |
| TAGnet [24] | 0.651 | 0.692 |
| IMSAT [25] | - | **0.984** |
| Aljalbout *et al.* [1] | 0.923 | 0.961 |
| MIXAE [3] | - | 0.945 |
| Spectral clustering [26] | 0.754 | 0.717 |
| SpectralNet [26] | 0.924 | 0.971 |
| ClusterGAN [21] | 0.89 | 0.95 |
| Info-GAN [27] | 0.86 | 0.89 |
| GAN with bp [21] | 0.90 | 0.95 |
| *MoE-Sim-VAE* (proposed) | **0.935** | 0.975 |

We compare the models with the Normalized Mutual Information (NMI) criterion but also classification accuracy (ACC) (Table 1). The MoE-Sim-VAE outperforms the other methods w.r.t. clustering performance when comparing NMI and achieves the second-best result when comparing ACC. Note that for comparability reasons we used the number of experts $k = 10$ in our model to fit the existing number of digits in MNIST. To prove that MoE-Sim-VAE is able to learn the correct number of experts, we report a study on synthetic data in supporting information (S1 Text and S1 Fig).

We use a UMAP projection [10] of MNIST as our similarity measure and then apply k-nearest-neighbors of each sample in a batch. In an ablation study, we show the importance of the similarity matrix to create a clear separation of the different digits in the latent representation. Therefore, we computed a test statistic based on the Maximum Mean Discrepancy (MMD) [28, 29] which can be used to test if two samples are drawn from the same distribution (see Section 1.2 in S1 Text). In this work we use MMD to test if samples of different clusters of the latent representation are similar. When sampling twice from the same cluster we get an average MMD test statistic of $t_{sim} = -0.05$ with, and $t = -0.11$ without similarity matrix, whereas the average distance between samples from two different clusters is significantly larger when training with similarity matrix $t_{sim} = 221.66$ compared to when training without $t = 49.29$. This clearly suggests better separation on the latent representations between the clusters when being able to define a respective similarity (S2 Fig).

In addition to the clustering network, we can make use of the latent representation for image generation purposes. The latent representation is trained as a mixture of standard Gaussians. The means of these Gaussians are the centers of the clusters trained via the clustering network. Therefore, the variational distribution can be sampled from and gated to the cluster-specific expert in the MoE-decoder. The expert then generates new data points for the specific data mode. Results and the schematic are displayed in Fig 2.

In an ablation study, we compare the two models MoE-Sim-VAE and VaDE [7] on generating MNIST images with the request for a specific digit. The goal is to show that a MoE decoder, as proposed in our model, is beneficial. We focus our comparison to VaDE since this model, as the MoE-Sim-VAE, resorts to a mixture of Gaussian latent representation but differs in generating images by means of a single decoder network instead of a Mixture-of-Expert decoder network. The rationale for our design choice is to ensure that smaller sub-networks learn to reproduce and generate specific modes of the data, in this case of specific MNIST digits.

To show that both models' latent representations are separating the different clusters well, we computed the Maximum Mean Discrepancy (MMD) [28], similar as introduced above. An MMD statistic of $t_{\text{MoE-Sim-VAE}} = 256.31$ and $t_{\text{VaDE}} = 355.14$ suggests separation of the clusters when sampling in the latent representations of both models. Therefore, both latent representations can separate the clusters of respective digits well, such that the decoder gets well-defined samples to generate the requested digit. Hence, the main difference of generating specific digits arises in the decoder/generator networks (S3 Fig).

We evaluated the importance of the MoE-Decoder to (1) accurately generate requested digits and (2) be efficient in generating requested digits. Specifically, we sampled 10, 000 points from each mixture component in the latent representation, generated images, and used the model's internal clustering to assign a probability to which digits were generated. To generate correct and high-quality images with VaDE, the posterior of the latent representation needs to be evaluated for each sample. This was done for the different thresholds $\phi \in [0.0, 0.1, 0.2, \cdots,$ 0.9, 0.999]. The default threshold [7] used was $\phi = 0.999$. To compare the separation of the clusters in the latent representation above using MMD, we used a threshold of only $\phi = 0.8$, which already is enough to have higher separation based on MMD. Instead of thresholding the latent representation, we ran the generation process for MoE-Sim-VAE for each threshold with the same settings. To generate images from VaDE we used the Python implementation (https://github.com/slim1017/VaDE) and model weights publicly available from Jiang *et al.* [7].

As a result the MoE-Sim-VAE generates digits more accurately with fewer resources required, especially when comparing the number of iterations required to fulfill the default posterior threshold of 0.999. VaDE needs nearly 2 million iterations to find samples that fulfill the aforementioned threshold criterion whereas the MoE-Sim-VAE only requires 10, 000 for a comparable sample accuracy. In comparison, the mean accuracy over all thresholds for MoE-Sim-VAE is 0.970, whereas VaDE reaches on average only 0.944 (S4, S5 and S6 Figs).

## Clustering organ-specific single cell RNA-seq data

Single-cell RNA-sequencing (scRNA-seq) measurements allow measuring transcriptomes of tens of thousands of single cells. Clustering of the resulting data into groups representing biological phenotypes, such as cell type or tissue type, constitutes a major analysis task in scRNA-seq studies. In the following, we present how MoE-Sim-VAE outperforms the methods Gaussian Mixture Models (GMM), k-means, hierarchical clustering, HDBSCAN, fuzzy-c-means (FCM), Louvain and scVI. [30–33] for clustering the scRNA-seq data of the Tabula Muris study covering seven different mouse organs [34]. The method scVI is a well established deep generative modeling framework designed for single cell transcriptomic data modeling the count data with a Poisson distribution, and allows to perform several downstream analysis tasks, such as clustering. Also in this example we used MoE-Sim-VAE with a BCE loss instead of a mean squared error loss for MoE-Sim-VAE, motivated due to better convergence properties in combination with artificial neural networks [15], and due to literature where BCE was used as reconstruction loss on RNA-seq data for visualization purposes [35].

MoE-Sim-VAE allows for incorporation of a user defined similarity and therefore also prior knowledge about the data. From literature we identified for each organ a representing signature gene and use each to encode prior expectation of organ assignment for each cell measurement. Namely, we take advantage of the high expressions of *Lpl* in the heart, *Miox* in kidney, *Hpx* in liver, *Tspan1* in large intestine, *Prx* in lung, *Cd79a* in spleen and *Dntt* in thymus [36–39]. Single cells which show above average expression of a respective signature gene are considered to be similar. For the training of MoE-Sim-VAE we only considered cells which show above average expression in exactly one of the respective organ-specific signature genes (does not apply for the test data). To highlight the influence of the similarity prior in this example, we report the average accuracy of 0.92 of having a correct similarity assignment for each organ based on above average expression (S1 Table).

MoE-Sim-VAE outperforms all above mentioned baseline approaches in clustering the single cells with respect to the organ of origin. Our model reaches a F-measure of 0.748 and is therefore close to 0.1 better compared to the second best competitor. We performed a hyperparameter screening for the competitor methods (more details in Chapter 4 in S1 Text) and chose the best results achieved on the test dataset based on the F-measure as well. In Table 2 we present the exact results in detailed comparison. In Fig 3A and 3B) we show a Principal Component Analysis (PCA) of the original data as well as of the latent representation of MoE-Sim-VAE. It can be seen that the organs are better separated in the latent representation inferred from our model which enables for better clustering results of MoE-Sim-VAE (Fig 3C)). In Fig 3D–3I) we present the results of the competitor methods in the latent representation of MoE-Sim-VAE and can clearly see that Louvain performs second best, but poorly separates cells from organs which are close to each other or overlap in the original PCA representation. Fig 3J) visualizes the Leiden clustering results on the latent representation inferred from scVI. This latent space shows a more detailed separated latent representation but with overlapping or separated true labels. For example, samples from the heart are separated in up to seven different groups. This leads to less precise clustering results concerning the task of identifying tissue types, but might be beneficial when clustering cell types. This also highlights

**Table 2. Results on clustering mouse organs based on RNA-seq.** We compare MoE-Sim-VAE to the competitor methods Gaussian Mixture Models (GMM), k-means, Hierarchical clustering, DBSCAN, FCM and Louvain clustering.

| Method | F-measure | NMI |
|---|---|---|
| GMM<br>PCA k = 20 | 0.632 | 0.487 |
| k-means<br>PCA k = 40 | 0.606 | 0.443 |
| hierarchical<br>PCA k = 20 | 0.643 | 0.534 |
| HDBSCAN<br>PCA k = 20<br>min cluster size = 50 | 0.615 | 0.517 |
| fuzzy-c-means<br>PCA k = 50<br>m = 4 | 0.549 | 0.336 |
| Louvain<br>PCA k = 30<br>resolution = 0.01 | 0.679 | **0.584** |
| scVI<br>leiden clustering<br>resolution = 0.06 | 0.653 | 0.561 |
| *MoE-Sim-VAE* (proposed) | **0.748** | 0.519 |

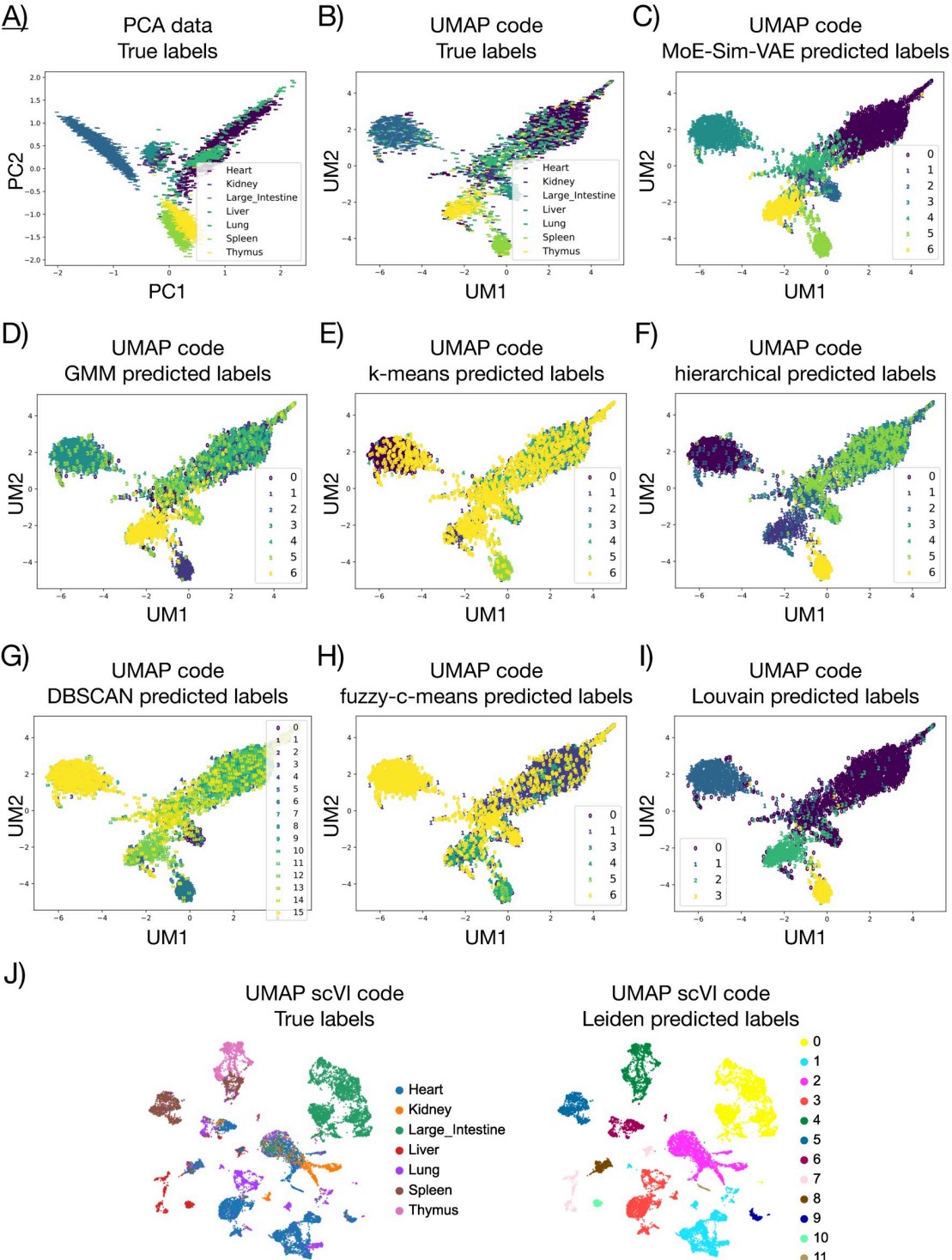

**Fig 3. Results of clustering mouse organs from single-cell RNA-sequencing data.** A) Principal Component Analysis of the original data with true labels. The remaining panels are UMAP representations of the latent representation inferred from MoE-Sim-VAE with B) true labels. C) predicted labels from MoE-Sim-VAE. D) predicted labels from Gaussian Mixture Model. E) predicted labels from k-means. F) predicted labels from hierarchical clustering. G) predicted labels from DBSCAN. H) predicted labels from fuzzy-c-means. I) predicted labels from Louvain. J) true and predicted labels on scVI inferred latent representation using Leiden clustering.

the importance of being able to incorporate prior knowledge when inferring latent representations for specific clustering tasks, such as grouping tissue types.

## Learning cell type composition in peripheral blood mononuclear cells using CyTOF measurements

In the following, we want to assess representation learning performance on the real-world problem of cell type definition from single-cell measurements. Cytometry by time-of-flight mass spectrometry (CyTOF) is a state-of-the-art technique allowing measurements of up to 1, 000 cells per second and in parallel over 40 different protein markers of the cells [40]. Defining biologically relevant cell subpopulations by clustering this data is a common learning task [41, 42].

Many methods have been developed to tackle the problem introduced above and were compared on four publicly available datasets in Weber and Robinson [42]. The best out of 18 methods were FlowSOM [43], PhenoGraph [44] and X-shift [45]. These are based on k-nearest-neighbors heuristics, either defined from a spanning graph or from estimating the data density. In contrast to these methods, MoE-Sim-VAE can map new cells into the latent representation, assign probabilities for cell types, and infer an interpretable latent representation, allowing intuitive downstream analysis by domain experts.

We applied MoE-Sim-VAE to the same datasets as in Weber and Robinson [42] and achieve superior results in classification using the F-measure [41] in three out of four datasets. Similarly as in Weber and Robinson [42], we trained MoE-Sim-VAE 30 times and report in Table 3 (adopted from Weber and Robinson [42]) the means and standard deviation across all runs (S7 Fig). As a MoE-Sim-VAE similarity measure we used a UMAP projection with Canberra distance [46] as metric and computed similarly to the MNIST experiments the k-nearest-neighbors of each sample in the batch. This applies for all CyTOF experiments.

**Table 3. Comparison of MoE-Sim-VAE performance to competitor methods in defining cell type composition in CyTOF measurements.** The results in the table are extracted from the review paper of [42], where 18 methods are compared on four different datasets. Our model outperforms the baselines on three out of four data sets.

| Method | Levine_32dim | Levine_13dim | Samusik_01 | Samusik_all |
|---|---|---|---|---|
| ACCENSE | 0.494 | 0.358 | 0.517 | 0.502 |
| ClusterX | 0.682 | 0.474 | 0.571 | 0.603 |
| DensVM | 0.66 | 0.448 | 0.239 | 0.496 |
| FLOCK | 0.727 | 0.379 | 0.608 | 0.631 |
| flowClust | N/A | 0.416 | 0.612 | 0.61 |
| flowMeans | 0.769 | 0.518 | 0.625 | 0.653 |
| flowMerge | N/A | 0.247 | 0.452 | 0.341 |
| flowPeaks | 0.237 | 0.215 | 0.058 | 0.323 |
| FlowSOM | **0.78** | 0.495 | 0.707 | 0.702 |
| FlowSOM_pre | 0.502 | 0.422 | 0.583 | 0.528 |
| immunoClust | 0.413 | 0.308 | 0.552 | 0.523 |
| k-means | 0.42 | 0.435 | 0.65 | 0.59 |
| PhenoGraph | 0.563 | 0.468 | 0.671 | 0.653 |
| Rclusterpp | 0.605 | 0.465 | 0.637 | 0.613 |
| SamSPECTRAL | 0.512 | 0.253 | 0.263 | 0.138 |
| SPADE | N/A | 0.127 | 0.169 | 0.13 |
| SWIFT | 0.177 | 0.179 | 0.202 | 0.208 |
| X-Shift | 0.691 | 0.47 | 0.679 | 0.657 |
| *MoE-Sim-VAE* (proposed) | 0.70 ± 0.04 | **0.68 ± 0.01** | **0.76 ± 0.03** | **0.74 ± 0.02** |

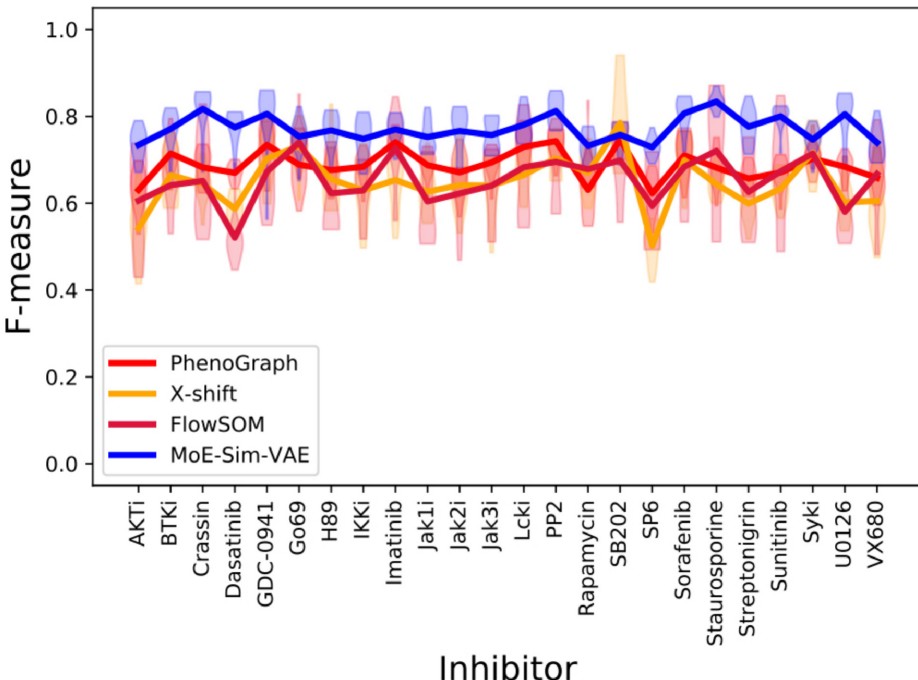

**Fig 4. Results on clustering cell types on CyTOF measurements.** Comparison of MoE-Sim-VAE to the most popular competitor methods on defining cell types in peripheral blood mononuclear cell data via CyTOF measurements. On the x-axis different inhibitor treatments are listed whereas the y-axis reports the respective F-measure. Each violin plot represents a run on a different inhibitor with multiple wells, whereas the line connects the means of the performance on the specific inhibitor.

Further, we trained a MoE-Sim-VAE model with a fixed number of experts *k* = 15 (thereby slightly overestimating the true number of subpopulations) on 268 datasets from Bodenmiller *et al.* [47] and achieve superior clustering results of cell subpopulations in the data when comparing to state-of-the-art methods in this field (PhenoGraph, X-Shift, FlowSOM). Results are summarized in Fig 4 as well as exactly listed in S2 Table. Furthermore, we visualize in detail the reconstruction of the original data per expert using a Principal Component Analysis on the original data space. This visualization also shows that many experts were silenced during the training, since only seven out of possible 15 experts where selected (S8 Fig).

## Discussion

Our MoE-Sim-VAE model can infer similarity-based representations, perform clustering tasks, and efficiently as well as accurately generate high-dimensional data. The training of the model is performed by optimizing a joint objective function consisting of data reconstruction, clustering, and KL loss, where the latter regularizes the latent representation. On the benchmark dataset of MNIST, we present superior clustering performance and the efficiency and accuracy of MoE-Sim-VAE in generating high-dimensional data. On the biological real-world tasks of clustering mouse organs and defining cell subpopulations in complex single-cell data, we show superior performances compared to state-of-the-art methods on a vast range of over 270 datasets and therefore demonstrate the MoE-Sim-VAE's real-world usefulness.

To achieve outstanding clustering performances the choice of the similarity measure as well as the hyperparameter tuning, such as for the loss coefficients, play a crucial role. As shown in

the ablation study for clustering MNIST, setting the similarity clustering loss to zero has a tremendous effect on the learned latent representation and the clustering performance. In general, we could observe that the loss coefficients of the reconstruction loss and clustering loss need to be selected close to one, whereas the loss coefficient for the KL loss is closer to zero. A less crucial role plays the selection of number of experts, as shown on clustering synthetic data, on the example of clustering mouse organs based on single cell RNA-sequencing data, or when clustering cell types from mass cytometry measurements. Even when defining more experts than the number of expected clusters, MoE-Sim-VAE did not target each single expert. A minimum number of required experts to distribute the different modes of the data with respect to the defined similarity where selected by the model.

Future work might include to add adversarial training to the MoE decoder, which could improve image generation to create even more realistic images. Also, specific applications might benefit from replacing the Gaussian with a different mixture model. Especially biological data is not always generated from Gaussian distributions. So far the MoE-Sim-VAE's similarity measure has to be defined by the user. Relaxing this requirement and allowing for learning a useful similarity measure automatically for inferring latent representations will be an interesting extension to explore. This could be useful in a weakly-supervised setting, which often occurs for example in clinical data consisting of healthy and diseased patients. Minor details between a healthy and diseased patient might make a huge difference and could be learned from the data using neural networks.

In summary, we expect the MoE-VAE model, as well its future extensions, to be a valuable contribution to the computational biology toolbox to identify biological group structure in high-dimensional molecular data modalities under consideration of weak prior knowledge, in particular including single-cell omics data.

## Supporting information

**S1 Text. Supporting information for Mixture-of-Experts Variational Autoencoder for clustering and generating from similarity-based representations on single cell data.**
(ZIP)

**S1 Fig. Testing MoE-Sim-VAE on data sampled from a Gaussian mixture model with randomly sampled parameters.** We tested for specific number of synthetic mixture components and iterating number of experts. Until a number of GMM components of23 MoE-Sim-VAE is precise in learning the real number of clusters even when allowing the model to have 40 experts.
(EPS)

**S2 Fig. Ablation study on the similarity matrix S.** Both figures show the MMD statistic and UMAP [10] projection of reconstructed MNIST digits computed on the latent representation. A) shows the results on MoE-Sim-VAE trained with the similarity matrix. The different digits separate well which can also be seen in the heatmap showing the MMD statistics between all digits. In comparison, B) shows results of the MoE-Sim-VAE model ignoring the similarity matrix setting the loss coefficient to zero. One can observe that the MMD statistic, which can be seen as a similarity measure of two distributions, is way lower compared to the model including the similarity matrix. Further, also the UMAP projection confirms less separation in the latent representation between the different digits.
(EPS)

**S3 Fig. Comparison of two sample MMD test on the distributions from the different mixture components in the latent representation.** The heatmaps on the left side show the

estimation of the MMD which can be seen as the distance between pairs of distributions. The figures on the right side show the separation of the cluster in the latent representation based on a dimensionality reduction via UMAP [10]. A) shows the results for the clusters of VaDE at a posterior threshold of 0.8 which is the first threshold which shows total separation of all clusters. B) shows the separation of the clusters in latent space learned from MoE-Sim-VAE. For both methods, all distributions belonging to clusters of different respective digits show a larger distance compared to the diagonal of matching distributions, such that we generate images from a well-separated latent representation for both methods and therefore the main difference comes from the decoders.
(EPS)

**S4 Fig. Comparison of data generation process between Moe-Sim-VAE and VaDE.** A) shows the accuracy of how certain a specific digit can be generated from the respective cluster in the latent representation whereas B) compares the number of runs until a sample from the latent representation satisfied the posterior criterion from VaDE. It needs to be mentioned that MoE-Sim-VAE does not require any thresholding such that we ran the data generation process multiple times with the same settings to compare with VaDE. In total 10000 samples are generated for each digit.
(EPS)

**S5 Fig. Confusion map for data generation using MoE-Sim-VAE.** Besides the systematic error of confusing digit 5 and 8, which can also depend on the clustering network, the digit generation of our model performs very precise with a high accuracy of generating the digit asked for. In comparison to VaDE [7] our model does not need any threshold on samples from the latent representation which reduces the computational costs by far.
(EPS)

**S6 Fig. Confusion maps for data generation using VaDE.** A) Posterior threshold 0.0. B) Posterior threshold 0.1. C) Posterior threshold 0.2. D) Posterior threshold 0.3. E) Posterior threshold 0.4. F) Posterior threshold 0.5. G) Posterior threshold 0.6. H) Posterior threshold 0.7. I) Posterior threshold 0.8. J) Posterior threshold 0.9. K) Posterior threshold 0.999 (default for VaDE [7]).
(EPS)

**S7 Fig. Reproducibility of MoE-Sim-VAE on the four datasets.** Similar as in Weter *et al.* [42], we show the reproducibility of MoE-Sim-VAE on the four datasets when running MoE-Sim-VAE 30 times. The variance on defining the correct subpopulations of MoE-Sim-VAE is quite small and therefore also an improvement to many methods compared in Weber *et al.* [42].
(EPS)

**S8 Fig. Reconstruction of data modes per expert.** PCA plot showing the reconstruction (red) of original data (colored underneath) separated per MoE-expert on the Inhibitor GDC-0941 and Well A09 from the Bodenmiller [47] data. This example reached a F-measure of 0.8606. The experts with ID 2, 3, . . ., 9 where not selected via the gating network. The red samples in each plot visualize the reconstructed data. A) Expert ID = 0. B) Expert ID = 1. C) Expert ID = 10. D) Expert ID = 11. E) Expert ID = 12. F) Expert ID = 13. G) Expert ID = 14. H) Visualization of the reconstruction taking the data modes from all selected experts together. I) PCA plot of the true labels without any reconstruction overlaid.
(EPS)

**S1 Table. Signature gene accuracy.** Accuracy of assigning a organ similarity based on high gene expression of prior selected organ specific signature genes for the split training and test data set. We computed the balanced accuracy for each single organ vs. the rest, respectively. (XLS)

**S2 Table. Exact results on 268 mass cytometry experiments.** CyTOF measurements from peripheral blood mononuclear cells (PBMCs) were taken and the goal is to define the different cell types present in the data. The ground truth was defined using the SPADE algorithm [48], which can visualize the high dimensional data in such a way to be able to manual gate the cells. We compare to other fully unsupervised methods as FlowSOM, X-shift and PhenoGraph and achieve in most cases the best F-measure. (XLS)

## Acknowledgments

AK thanks Florian Buettner for helpful discussions and his inspirational attitude.

## Author Contributions

**Conceptualization:** Andreas Kopf, Vincent Fortuin, Manfred Claassen.

**Funding acquisition:** Manfred Claassen.

**Investigation:** Andreas Kopf, Vignesh Ram Somnath.

**Project administration:** Manfred Claassen.

**Resources:** Manfred Claassen.

**Supervision:** Manfred Claassen.

**Writing – original draft:** Andreas Kopf, Manfred Claassen.

**Writing – review & editing:** Andreas Kopf, Vincent Fortuin, Manfred Claassen.

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
