## [Decision Letter · Decision Letter 0]

22 Mar 2021

Dear Prof. Dr. Claassen,

Thank you very much for submitting your manuscript "Mixture-of-Experts Variational Autoencoder for Clustering and Generating from Similarity-Based Representations on Single Cell Data" for consideration at PLOS Computational Biology.

As with all papers reviewed by the journal, your manuscript was reviewed by members of the editorial board and by several independent reviewers. In light of the reviews (below this email), we would like to invite the resubmission of a significantly-revised version that takes into account the reviewers' comments.

We cannot make any decision about publication until we have seen the revised manuscript and your response to the reviewers' comments. Your revised manuscript is also likely to be sent to reviewers for further evaluation.

Sincerely,

Qing Nie

Associate Editor

PLOS Computational Biology

Alice McHardy

Deputy Editor

PLOS Computational Biology

Reviewer's Responses to Questions

**Comments to the Authors:**

Reviewer #1: The manuscript `Mixture-of-Experts Variational Autoencoder for Clustering and Generating from Similarity-Based Representations on Single Cell Data` proposes a generative clustering model, which is based on a variational autoencoder with a Gaussian mixture model for the latent space and a decoder consisting of a mixture of networks, where each mode in the latent space is decoded by an expert network. The gating network to assign samples to modes can be guided by prior knowledge on sample similarity. The authors demonstrate the performance of their model on MNIST data as well as single cell RNA-seq and mass cytometry clustering tasks. Overall, the work is interesting and the experiments conducted in a thorough manner. For an added value to the community, the authors should make their methods and scripts to reproduce the results accessible and provide additional details on their experiments. Other points are listed below.

Major points:

- Please make the code for the method and the scripts to reproduce the experiments accessible.

- What pre-processing was applied to the data presented in the Results? Please specify this and also provide details on the split of train and test data.

- - Several methods exist that use non-Gaussian variational distribution in a VAE, especially count based models in the single cell domain. These should be mentioned in p.2 l.14 and the results of a clustering in their latent space included in Table 2.

- The ablation study nicely highlights the benefits of including a similarity matrix. However, the exact impact of the chosen similarity metric remains unclear. For this, a clustering based solely on the chosen similarity matrix should be included as a reference in all experimental results. Does the influence of the similarity matrix depend on the dimensions of the data or number of clusters? How sensitive is the method to misspecifications in the similarity (e.g. wrongly chosen signature genes in the cell type clustering problem)?

Minor points:

- Could the authors comment on the motivation for the binary cross-entropy as reconstruction loss in Eq. (4). This seems to lead to blurry images in Fig. 2 and it is unclear why this should be a suitable loss for RNA-seq data.

- What would be the guidelines to choose the number of clusters K in applications? This seems to be crucial for the ability to generate samples and according to Fig S1 many more experts than actual clusters in the data might be found.

- Table 1 would be more insightful if the authors could provide a short description for these methods. Also, GANs for clustering seem an important alternative for the task but are not contained in the comparison nor mentioned in the text.

- The ablation study could be included as part of Table 1.

- l117-136 are hard to follow without going back to the original publication of VaDE.

- The authors could remove repetitive parts of the text (e.g. method description of VaDE and MoE-Sim-VAE in Results, eq (2) and (3)) and better separate general concepts and technical details in the introduction.

- How sensitive is the method to the choice of pi_1 and pi_2?

Reviewer #2: The authors report a computational method, Mixture-of-Experts Similarity Variational Autoencoder, at clustering and data generation, with applications on large-scale single-cell data. The proposed mathematical framework is solid and builds upon ideas that are appropriate for the analysis of large-scale single-cell data. The authors have demonstrated the applicability of their method using publicly available data sets, and the figures and tables are simple and clear. I however have a few suggestions to improve the manuscript.

Major comments:

1. The authors do not a link to their implementation. This should be a red line for academic computational tools, and I would request the authors to share their implementation via a github repository reproducible in a revised version of the manuscript. In addition, the authors should include a vignette or a tutorial reproducing the results from at least one of the applications presented in the manuscript.

2. In Table 2 the authors extract the F-measure and NMI scores from a public review (Lukas et al. 2016). This is fine as long as the processing pipelines that the author’s followed is the same as in the review. Minor changes in the data processing pipelines can lead to significant differences in the results. If this is the case, the authors should clarify. If not, the authors should verify the reported results with their own data processing pipelines.

Minor comments:

1. Fig3 is missing the colour legend

2. Could the authors back the quantitative results shown in Fig4 with a visualisation of the data, with cells coloured by cluster / cellt ype?

3. Why do the authors use the binary cross entropy as a reconstruction loss instead of the mean-squared error? Single-cell data is usually not scaled between 0 and 1.

**Have all data underlying the figures and results presented in the manuscript been provided?**

Reviewer #1: **No: **Code to reproduce the results is missing.

Reviewer #2: Yes

PLOS authors have the option to publish the peer review history of their article (what does this mean?). If published, this will include your full peer review and any attached files.

Reviewer #1: No

Reviewer #2: No
---

## [Decision Letter · Decision Letter 1]

14 May 2021

Dear Prof. Dr. Claassen,

We are pleased to inform you that your manuscript 'Mixture-of-Experts Variational Autoencoder for Clustering and Generating from Similarity-Based Representations on Single Cell Data' has been provisionally accepted for publication in PLOS Computational Biology.

The reviewer one has two minor comments. Please address. 

Best regards,

Qing Nie

Associate Editor

PLOS Computational Biology

Alice McHardy

Deputy Editor

PLOS Computational Biology

Please address Reviewer 1's two minor comments in the final version of the submission.

Reviewer's Responses to Questions

**Comments to the Authors:**

Reviewer #1: The authors have addressed my previous comments in their revised manuscript.

Before publication they should still address the following minor points:

1. Please fix typos and sentence structure in lines 66/67 and 261.

2. Please complete the code repository to make all examples from the paper reproducible and include the code that was used for the comparison to other methods in the benchmarks.

Reviewer #2: The authors have correctly addressed all my comments. I recommend publication of this manuscript.

**Have the authors made all data and (if applicable) computational code underlying the findings in their manuscript fully available?**

Reviewer #1: **No: **see comments to the Authors

Reviewer #2: None

PLOS authors have the option to publish the peer review history of their article (what does this mean?). If published, this will include your full peer review and any attached files.

Reviewer #1: No

Reviewer #2: No

---

## [Editor Report · Acceptance letter]

25 Jun 2021

PCOMPBIOL-D-20-02250R1 

Mixture-of-Experts Variational Autoencoder for Clustering and Generating from Similarity-Based Representations on Single Cell Data

Dear Dr Claassen,

I am pleased to inform you that your manuscript has been formally accepted for publication in PLOS Computational Biology. Your manuscript is now with our production department and you will be notified of the publication date in due course.

With kind regards,

Katalin Szabo
